# Urinary Metabolomics Validates Metabolic Differentiation Between Renal Cell Carcinoma Stages and Reveals a Unique Metabolic Profile for Oncocytomas

**DOI:** 10.3390/metabo9080155

**Published:** 2019-07-24

**Authors:** Oluyemi S. Falegan, Shanna A. Arnold Egloff, Andries Zijlstra, M. Eric Hyndman, Hans J. Vogel

**Affiliations:** 1Department of Biological Sciences, University of Calgary, Calgary, AB T2N 4V8, Canada; 2Department of Pathology, Microbiology, and Immunology, Vanderbilt University Medical Center, Vanderbilt University, Nashville, TN 37232, USA; 3Department of Veterans Affairs, Nashville, TN 37212, USA; 4Department of Surgery, Cumming School of Medicine, University of Calgary, Calgary, AB T2N 4N1, Canada; 5Prostate Cancer Centre, Rockyview Hospital, Calgary, AB T2V 1P9, Canada; 6Department of Biochemistry and Molecular Biology, Cumming School of Medicine, University of Calgary, Calgary, AB T2N 4N1, Canada

**Keywords:** Metabolomics, Renal cell carcinoma, Nuclear Magnetic Resonance, Oncocytoma, Histology

## Abstract

Renal cell carcinoma (RCC) is a heterogeneous malignancy which often develops and progresses asymptomatically. Benign oncocytomas are morphologically similar to malignant chromophobe RCC and distinguishing between these two forms on cross-sectional imaging remains a challenge. Therefore, RCC-specific biomarkers are urgently required for accurate and non-invasive, pre-surgical diagnosis of benign lesions. We have previously shown that dysregulation in glycolytic and tricarboxylic acid cycle intermediates can distinguish benign lesions from RCC in a stage-specific manner. In this study, preoperative fasting urine samples from patients with renal masses were assessed by ¹H nuclear magnetic resonance (NMR). Significant alterations in levels of tricarboxylic acid cycle intermediates, carnitines and its derivatives were detected in RCC relative to benign masses and in oncocytomas vs. chromophobe RCC. Orthogonal Partial Least Square Discriminant Analysis plots confirmed stage discrimination between benign vs. pT1 (R2 = 0.42, Q2 = 0.27) and benign vs. pT3 (R2 = 0.48, Q2 = 0.32) and showed separation for oncocytomas vs. chromophobe RCC (R2 = 0.81, Q2 = 0.57) and oncocytomas vs. clear cell RCC (R2 = 0.32, Q2 = 0.20). This study validates our previously described metabolic profile distinguishing benign tumors from RCC and presents a novel metabolic signature for oncocytomas which may be exploited for diagnosis before cross-sectional imaging.

## 1. Introduction

Kidney cancer is responsible for approximately 4.2% of all cancer cases reported in the United States in 2019 [1]. Renal cell carcinoma (RCC) is the most common form of kidney cancer (85 %) and is considered the most lethal genitourinary cancer due to its high mortality rate [2]. RCC is also a heterogeneous disease culminating in different histological sub-types which feature distinct morphological, genetic and metabolic differences. Benign renal masses, known to be indolent and rarely progress to malignancy, are frequently encountered in medical practice. While many benign renal lesions have clear radiographic features that distinguish them from RCC, the common benign tumors: oncocytomas (ONC) and angiomyolipomas (AMLs) are more difficult to differentiate from RCC with current imaging techniques because they share some morphological and histological resemblances.

Distinguishing oncocytomas from chromophobe and eosinophilic RCC poses a major clinical challenge due to similarities in the demographics of the presenting patient population, asymptomatic nature and growth rate [3]. On the other hand, AMLs can be diagnosed by cross-sectional imaging with a high degree of accuracy due to unique features, such as smooth muscles, aneurysmal blood vessels and adipose tissue. However, AMLs can be confused with fat-retaining RCCs and fat-lacking AMLs may be missed or erroneously diagnosed as RCC, leading to overtreatment and unnecessary morbidity [4].

Heterogeneity in RCC stems from genetic alterations, the most common RCC subtype, clear cell renal cell carcinoma (ccRCC), is characterized by alterations in the *Von Hippel Lindau* (*VHL*) gene which subsequently impacts downstream metabolic processes such as cellular glucose transport. The papillary RCC (PRCC) subtype is associated with mutations in the *fumarase hydratase* gene, where the function of the resultant and defective tricarboxylic acid (TCA) cycle enzyme fumarase is inhibited [5]. Alterations in genes (*SDHB*) encoding yet another TCA cycle enzyme, succinate dehydrogenase, have been reported in patients with hereditary paragangliomatosis with phaeochromocytomas and in some ccRCC cases [6]. Additionally, low expression of isocitrate dehydrogenase (IDH1) was recently reported to be associated with poor prognosis in ccRCC [7]. Isocitrate dehydrogenase catalyzes the conversion of the TCA intermediate isocitrate to α-ketoglutarate (alphaKG). Taken together, genetic mutations in RCC result in characteristic metabolic alterations which may be exploited for crucial diagnostic benefits.

^1^H nuclear magnetic resonance (^1^H NMR) metabolomics is a highly reproducible and non-destructive technique for evaluating a large complement of metabolites. It is useful for identifying metabolic alterations in body fluids and the resulting metabolic profile can distinguish between normal, benign and malignant lesions [8,9,10]. In our preliminary report, we applied ^1^H NMR and gas chromatography mass spectrometry (GCMS) based metabolomics analyses to urine and serum specimens to differentiate between benign and malignant renal masses in a small patient cohort (53 samples). In that study, we found that glycolytic and TCA cycle metabolites in blood and urine significantly separated benign lesions from ccRCC [11]. The present study is aimed at validating this previously identified urinary metabolic signature within a larger, external RCC cohort (Vanderbilt Cohort) in addition to exploring the metabolic differences between histologic subtypes of RCC. Identifying metabolic markers in biofluids that can distinguish between RCC histologic subtypes has the potential to improve screening and enable differential diagnosis prior to surgical resection. Metabolic markers also have the potential to enhance prognostication and disease staging of renal cell carcinoma. 

## 2. Results

### 2.1. ^1^H NMR Models Separate Benign Lesions from All RCC Stages

Overall, 145 ^1^H NMR spectra were collected and used for metabolic profiling of urine samples, described in Table 1. On average, 73 urine metabolites were identified and quantified for each sample.

An unsupervised PCA model was first built for the whole data set which showed no distinct separation between the (Benign, pT1, pT2, pT3 and pT4) groups (Figure 1a). Seven outliers (three of the pT1 group, two of the pT2 group and two of the pT3 group) which were samples located outside the Hotellings T2 ellipse of the PCA model were identified and excluded from further analyses (Figure 1b–e). Nevertheless, the R2Y and Q2 metrics were comparable before and after outlier exclusion.

To reveal the metabolic differences between groups, two sets of supervised Orthogonal Partial Least Square Discriminant Analysis (OPLS-DA) models were constructed, OPLS-DA focuses on creating a model that separates groups of observations on the basis of their x-variables (metabolites with VIP values > 1) (Figure 1).

Firstly, we observed separation between benign samples and pT1, pT2, pT3 and pT4 samples when all RCC histological subtypes were considered (Figure 1b–e). Benign versus pT1 (R2Y = 0.30: Q2 = 0.15) and benign versus pT3 (R2Y = 0.37: Q2 = 0.12) models showed poor separation with relatively low statistical metrics, while benign versus pT2 (R2Y = 0.75; Q2 = 0.64), and benign versus pT4 (R2Y = 0.97; Q2 = 0.67) comparisons showed better separation and improved metrics when all histology subtypes were included. Separation between benign lesions and RCC samples of all histology types did not show distinct separation (Appendix A
Figure A1).

Secondly, benign lesions were compared with samples of ccRCC histology, a comparison most closely related to the groups compared in our previous study in which 92.5% of the RCC samples analyzed were ccRCC [11]. Distinct group separations were confirmed (Figure 2) and the metrics were similar to previously reported data; whereby, benign versus pT1 (R2Y = 0.42; Q2 = 0.27; Figure 2a) and benign versus pT2 showed the best group separation without overlap between the groups (R2Y = 0.96; Q2 = 0.82; Figure 2b). On the other hand, there was some overlap between benign renal lesions versus pT3 samples (R2Y = 0.48; Q2 = 0.32; Figure 2c) and benign versus pT4 (R2Y = 0.83; Q2 = 0.65; Figure 2d).

Overall, the urine samples analyzed by NMR and OPLS-DA showed improved separation between benign and malignant groups in cases where benign lesions were compared with ccRCC histology samples. Statistical metrics are outlined in Table 2.

We next assessed the utility of NMR spectroscopy for distinguishing between other stages of renal cell carcinoma when all histologic subtypes were considered as well as, when only ccRCC was included. OPLS-DA models revealed relatively poor group separation between different RCC stages when all histology types were included in the analysis; pT1 vs pT2 (R2Y = 0.30; Q2 = 0.19), pT2 vs pT3 (R2Y = 0.41; Q2 = 0.26), pT1 vs pT4 (R2Y = 0.33; Q2 = 0.12) and pT3 vs pT4 (R2Y = 0.37; Q2 = 0.16). Separation between pT2 vs pT4 (R2Y = 0.74; Q2 = 0.42) provided the only statistically significant model. The OPLS-DA models created to examine group separation between cancer stages in samples assigned with ccRCC showed no feasible separation between the groups except in pT1 vs pT2 (R2Y = 0.52; Q2 = 0.29) and pT2 vs pT3 (R2Y = 0.41; Q2 = 0.16) (Appendix A
Table A1).

### 2.2. Group Separation between RCC Histology Subtypes

Given the similarities between some benign lesions and RCC, we wanted to identify possible metabolic alterations that may distinguish these benign lesions from the RCC histological subtypes represented in our study. Oncocytomas showed considerable separation from chromophobe (R2Y = 0.81; Q2 = 0.57; Figure 3a) and ccRCC (R2Y = 0.32 Q2 = 0.20; Figure 3b) (Table 2). However, AMLs showed no distinct metabolic profile when compared with all RCC subtypes. Also, models comparing between ccRCC, chromophobe and papillary RCC subtypes showed some separation between the groups but none were statistically significant.

### 2.3. Confirmed Differential Metabolites; Potential RCC Biomarkers

Metabolites contributing significantly to the separation between benign renal lesions and stages of RCC were identified and were similar to our previously identified repertoire (Table 3). Citrate and succinate specifically contributed to the differential separation. Also, a decrease in RCC glycine levels was confirmed for every group comparison. Increases in pyruvate and lactate levels were not statistically significant. However increased o-acetylcarnitine and carnitine were detected, with increased gluconate levels seen in benign vs pT2. Additionally, higher amounts of urinary methylhistidine, histamine, taurine and methionine contributed to the group separation in the benign vs pT3 and benign vs pT4 models.

### 2.4. Metabolic Distinction between Histological Subtypes

The model comparing oncocytomas to chromophobes (Figure 3a) showed distinct group separation which was attributed to reduced citrate and increased carnitine, trans-aconitate, succinate and π-methylhistidine in chromophobe RCC, while lower levels of citrate, 1-methylnicotinamide, glycine, 2-hydroxyisobutyrate and higher carnitine, tartrate, trans-aconitate and histamine in ccRCC relative to oncocytoma were observed (Table 3). This cohort consisted of only one sample of the collecting duct histology subtype which was excluded from the analysis.

## 3. Discussion

Most RCC patients are diagnosed incidentally and distinguishing benign renal masses from RCC on ultrasound (US) and cross-sectional imaging such as computed tomography (CT) and magnetic resonance imaging (MRI) are not always accurate [11]. In a previous study, we showed the potential of metabolomics analysis for distinguishing benign renal masses from stages pT1 and pT3 of RCC using non-invasive means; there were no samples of stages pT2 and pT4 available for similar comparisons in that cohort [12]. The premise of metabolomics is founded on detecting changes in cellular metabolic profiles that are induced by oncogenic processes. Measuring the changes of these metabolic products potentially allows for the identification and differentiation between malignant and benign tissue.

In the present study, we applied ^1^H NMR to validate our previously identified RCC-specific metabolic signature in urine and to assess the metabolic difference in distinguishing benign oncocytomas and angiomyolipomas from malignant renal cell carcinoma histologic subtypes. For these purposes, urine is an ideal biofluid for metabolomics studies of RCC due to its direct contact with the urinary system and ease of obtaining substantial sample volumes. Overall, this study confirms our previous report on the discriminatory power of ^1^H NMR coupled with multivariate statistical analysis (OPLS-DA) in separating benign lesions from pT1 and pT3 disease based on differential urinary citrate, glycine and succinate levels, and in addition shows metabolic difference between benign lesions and pT2 and pT4 RCC.

We confirmed group separation between benign lesions and all RCC stages and validated previously identified RCC metabolites. The prevalence of aerobic glycolysis is reiterated by significantly reduced TCA cycle metabolites: citrate and succinate in all RCC stages coupled with elevated but not significantly increased pyruvate and lactate levels. Downregulation of glycine in RCC is confirmed in this cohort, a finding which correlates with previous studies of urine of prostate cancer patients [13]. Glycine is an essential amino acid required by proliferating cancer cells for energetic purposes and performs a similar function as serine in sustaining the one-carbon metabolic pathway which supplies precursors for the biosynthesis of biomolecules essential for cancer cell growth [14]. Rapidly proliferating cancer cells have shown increased glycine-dependence, correlation between increased glycine consumption and rapid proliferation. Using consumption and release (CORE) analysis, rapidly proliferating LOX IMVI cells were shown to consume glycine and harness it for de novo biosynthesis of purine nucleotides [15].

Metabolic dysregulation in RCC, especially glucose metabolism is known to be differentially partitioned. Such that metabolites in the upper half of the glycolytic pathway and genes encoding glucose transporters are significantly increased, while intermediates in the lower half of the pathway and specific TCA cycle metabolites and genes are reduced in response to RCC [16,17]. This partitioning is attributed to the diversion of upper glycolytic intermediates towards the pentose phosphate pathway for the synthesis of ribose-5-phosphate and NADPH and the lower intermediate towards the TCA cycle or one-carbon metabolism [16]. In the current study, intermediates of the upper and lower glycolytic pathway such as glucose, G6P, and fructose 6-phosphate (F6P) and metabolites after F6P were absent, it is important to note that these metabolites were measured in tissue extracts in the studies reporting alterations, compared to our urine-based analysis. Also, glucose is metabolized in the human body and filtered in the glomeruli (about 180g/day), it would be detrimental to the human system to lose such enormous amount of glucose, so it is reabsorbed in the proximal tubule and this may explain the absence of glucose and other glycolytic intermediates in our urine samples [18].

Conversely, lower amounts of TCA cycle metabolites such as citrate and succinate but not malate and fumarate were detected in our urine samples. Citrate and succinate levels reduced throughout our analysis and in agreement with the findings of previous urine-based studies [19,20]. These observations are reflective of an impaired TCA cycle with possible impact on mitochondrial bioenergetics and oxidative phosphorylation.

In this study, we identified a distinct urinary metabolic signature which distinguishes benign oncocytomas from malignant chromophobes. Benign oncocytomas and malignant chromophobe RCC are both derived from intercalated cells of the collecting duct. Based on this similarity in origin, these variants share morphological features which pose a clinical challenge in accurately distinguishing them for pre-surgical diagnosis [21]. Several immunohistochemical and molecular markers are reported to distinguish between ONC and chromophobes; however, the clinical utility of these markers is limited [22,23].

We identified lower citrate and higher carnitine and trans-aconitate in chromophobes and ccRCC compared to oncocytomas. This agrees with our previous study where urinary citrate decreased in RCC relative to benign controls and downregulation of citrate in chromophobes may corroborate the prevalence of the RCC metabolic hallmark (i.e., Warburg effect), partly evidenced by reduced TCA-cycle intermediates.

Significantly higher urinary carnitine was detected in chromophobes and ccRCC relative to oncocytomas. This finding is particularly interesting as elevated levels of carnitine and its derivatives have been previously reported in the urine of RCC patients and confirmed in xenograft models and RCC cell lines [9,24]. These authors, however, did not define the histological class of the samples involved but revealed a stage-dependent increase of urinary carnitine and acylcarnitines in RCC. In addition, we have detected increased o-acetylcarnitine in all RCC stages relative to benign samples (Table 2). Carnitine and acylcarnitines are essential intermediates for the transfer of long chain fatty acids to the mitochondria for β-oxidation and an accumulation in the cell may imply an increased demand as a result of rapid fatty acid oxidation to meet higher tumoral energy requirements [24]. Conversely, Wettersten et al., found that the accumulation of acylcarnitine did not overlap with upregulation of fatty acid oxidation enzymes in RCC [25]. These authors suggested that, in this case, fatty acid β-oxidation may be downregulated as RCC progresses, with the resultant effect being the accumulation of unused acylcarnitines which may be used in non-energy related processes.

Taken together, carnitine and its derivatives, specifically o-acetylcarnitine, have a unique association with chromophobe and clear cell RCC histological subtypes as identified in our results and may point to elevated fatty acid oxidation in these RCC subtypes which differentiates them from benign oncocytomas. Be that as it may, the prevalence of β-oxidation in RCC remains to be confirmed, as conflicting results have thus far been reported [26].

While the present study confirmed the prevalence of metabolites that are a hallmark of aerobic glycolysis in RCC, this metabolic signature may be insufficient as biomarkers for overcoming the clinical dilemma that RCC heterogeneity brings. A metabolic panel that synergizes the confirmed metabolic signature and carnitine/derivatives which have shown considerable histology differentiation may have promising clinical applicability in differentiating benign from RCC lesions, specifically oncocytic tumors where imaging techniques fail.

We recognize that the ONC/Chromophobe model in our study is flawed by small sample number, which are fewer than required to achieve a statistically strong comparison. Given that benign lesions are not often encountered but stumbled upon during surgical intervention of renal cell carcinoma, this number of oncocytomas is typical in practice. Nevertheless, to ensure reduced bias in computing this model, we used a five-fold cross validation for the model calculation. A validation of the model separation and differential metabolite(s) associated with oncocytomas is required in future follow-up studies with larger sample size.

## 4. Materials and Methods

### 4.1. Patient Enrollment and Sample Collection

Ethics approval for this study was obtained from the Vanderbilt University Institutional Review Board (IRB# 140888) and the Institutional Research Information Services Solution (IRISS) of the University of Calgary. Ethical guidelines were followed in the conduct of the research. Urine samples were collected by Foley catheter at the time of nephrectomy but prior to incision from fasting patients. Urine samples were stored at −80 °C from 2013 to 2015, thawed in batches on ice to aliquot into 1.5 mL freezer tubes, and then again stored at −80 °C until usage/shipping. Benign lesions were determined by post-operative pathology and compared to pathologically confirmed RCC. We performed a case-control analysis on 135 malignant samples and 10 benign samples of urine from the same cohort of patients. The malignant groups were stratified by pathological stage pT1 (*n* = 59), pT2 (*n* = 24), pT3 (*n* = 45) and pT4 (*n* = 7). The clinicopathologic characteristics of the samples are shown in Table 1.

### 4.2. ^1^H NMR Spectral Collection

NMR analysis was performed on a 600 MHz Bruker Ultrashield Plus NMR spectrometer (Bruker BioSpin Ltd., Milton, ON, Canada) following procedures described previously [27,28]. Briefly, 145 urine samples were thawed on ice and 200 uL of each sample filtered in prewashed 3 kDa NanoSep microcentrifuge filters (Pall, Inc., East Hills, NY, USA) to remove protein and other large impurities. The filtrates were then transferred to clean microcentrifuge tubes and phosphate buffer, sodium azide and D_2_O were added. Untargeted one-dimensional proton ^1^H NMR analysis was carried out using the ‘noesygppr1d.2’ standard pulse program for improved water suppression [29]. The resulting spectra were manually processed (phasing, baseline correction, referencing to the DSS peak at 0.0 ppm) and profiled using the Processor and Profiler modules of the Chenomx NMR Suite 7.5 software (Chenomx Inc. Edmonton, Canada) respectively. Metabolites were detected and quantified using the Chenomx Suite reference libraries [30].

### 4.3. Multivariate Statistical Analysis

NMR data was normalized using the median fold change method [31]. Normalized data was used for multivariate statistical analysis in SIMCA-P+ 14.1 software (Umetrics, Umea, Sweden) where log transformation, centering and unit variance scaling were carried out [32,33,34]. All measured metabolites were used for further analysis using an untargeted and comparative approach. Unsupervised principal component analysis (PCA) models were initially constructed to identify potential outliers and groups of observations that may form distinct patterns, this was followed by generating supervised orthogonal partial least squares-discriminant analysis (OPLS-DA) statistical models in which two groups were compared per time after outliers were excluded [35]. These models were based on selected metabolites that had a Variable Influence on Projection (VIP) value greater than 1 [32,36,37]. In SIMCA, the presence of an orthogonal component in an OPLS-DA model determines the appearance of the score plot, a model calculated with more than one component (orthogonal and predictive components) is presented in a score-plot which shows observations bounded by an ellipsoid representing the 95% confidence interval of the Hotelling’s T-squared distribution. Observations that fall outside this ellipsoid are considered outliers. When an OPLS-DA model is calculated based on one predictive component, the visual plot is displayed in a score plot with the observations bounded by lines which represent the 2SD and 3SD limits. The variation (R2Y) and predictive ability (Q2) of the OPLS-DA models were calculated based on seven-fold cross-validation, except for the oncocytoma vs. chromophobe model where a k-fold cross validation (*k* = 5) was applied considering that the sample number was less than 20 [38]. Statistically significant OPLS-DA models were confirmed by the CV-ANOVA *p*-value (*p* < 0.05) and significantly different metabolites between classes were considered potential biomarkers. A 999 times permutation test was also conducted for OPLS-DA models in which negative Q2 intercepts were calculated [39]. In addition, the area under the receiver operating characteristics curve (AUC) was generated using the ROC tool in SIMCA-P+ 14.1 (Umetrics, Umea, Sweden).

To confirm the predictive ability and the validity of the OPLS-DA models generated in the previous study [12], separate OPLS-DA models were constructed based on samples from the current study using the same comparisons as before (B vs pT1 and B vs pT3).

## Figures and Tables

**Figure 1 metabolites-09-00155-f001:**
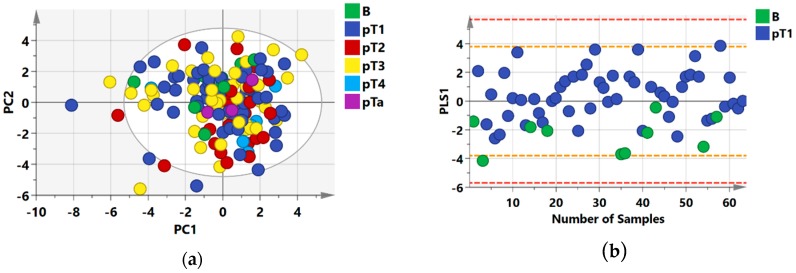
NMR Statistical Models showing separation between benign lesions and combined histological subtypes in supervised models (**a**) Principal Component Analysis; PCA-X scatter score plot, (**b**) Orthogonal Partial Least Squares Discriminant Analysis (OPLS-DA) scatter score plot benign versus stage 1 cancer cases; (**c**) benign versus stage 2 cancer cases (**d**) benign versus stage 3 cancer cases (**e**) benign versus stage 4 cancer cases; along their orthogonal partial least squares (OPLS1) and partial least squares components (PLS1). The white spheres in (**a**) and (**e**) describe the 95% confidence interval of the Hotelling’s T-squared distribution and the orange and red dashed lines in (**b**–**d**) describe the 2SD and 3SD limits respectively.

**Figure 2 metabolites-09-00155-f002:**
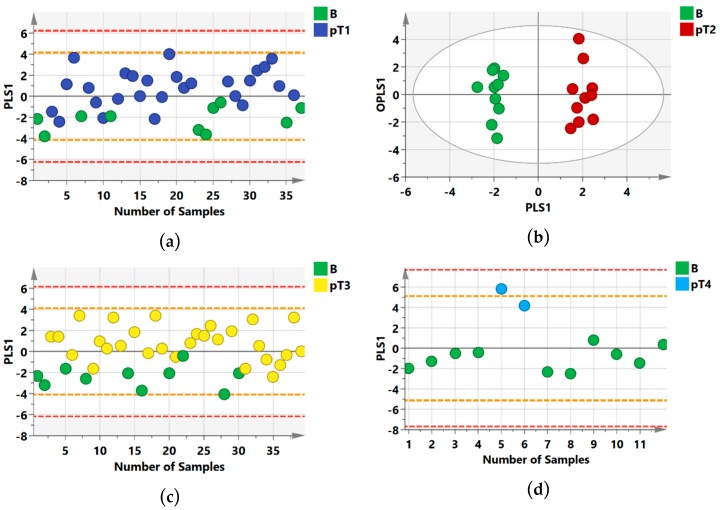
NMR Orthogonal Partial Last Squares Discriminant Analysis (OPLS-DA) score scatter plots distinguishing between benign and clear cell renal cell carcinoma. (**a**) benign versus stage 1 cancer cases; (**b**) benign versus stage 2 cancer cases (**c**) benign versus stage 3 cancer cases (**d**) benign versus stage 4 cancer cases along their orthogonal partial least squares (OPLS1) and partial least squares components (PLS1). The white spheres in (**b**) describe the 95% confidence interval of the Hotelling’s T-squared distribution and the orange and red dashed lines in (**a**,**c**,**d**) describe the 2SD and 3SD limits respectively.

**Figure 3 metabolites-09-00155-f003:**
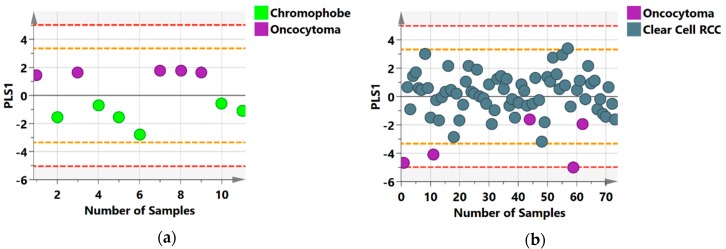
NMR Orthogonal Partial Last Squares Discriminant Analysis (OPLS-DA) score scatter plots distinguishing between histology subtypes (**a**) chromophobe versus oncocytomas cases; (**b**) oncocytomas versus ccRCC cases along their partial least squares component (PLS1). The orange and red dashed lines describe the 2SD and 3SD limits respectively.

**Table 1 metabolites-09-00155-t001:** Clinicopathologic characteristics of 145 patients with renal masses undergoing metabolomics analysis. RCC: Renal cell carcinoma.

Sample Group	Number of Samples	Age at Surgery (Range in Years)	Mean Age (Years)	Number of Men	Number of Women
**Benign Control**	10	47–89	63	6	4
Angiomyolipoma	5	47–89	66	2	3
Oncocytoma	5	55–73	61	4	1
**RCC**	134	35–99	61	96	38
ccRCC	74	35–99	59	51	23
Papillary	22	51–84	66	18	4
Chromophobe	6	53–68	61	3	3
RCC mixed type	4	53–70	61	2	2
Collecting duct carcinoma	1	62	-	-	1
Transitional cell carcinoma	27	44–82	65	22	5
Unclassified	1	35	-	-	1
Non-invasive papillary urothelial carcinoma (pTa)	4	50–66	58	3	1
Stage I (pT1)	59	35–99	63	41	18
Stage II (pT2)	24	35–84	58	17	7
Stage III (pT3)	45	37–82	62	30	15
Stage IV (pT4)	7	37–71	62	7	0
BMI 19–25	40	35–89	63	2	19
BMI above 25	102	35–81	61	79	23
BMI unknown	3	41–61	52	2	1

BMI: Body Mass Index. ccRCC: clear cell renal cell carcinoma. The bold categories indicate that the classes under them are subcategories of the ones in Bold.

**Table 2 metabolites-09-00155-t002:** Statistical metrics of group separation.

Model Type	R2Y	Q2	CV ANOVA *p*-Value	Q2 Intercept	Sensitivity	Specificity
Benign vs RCC (All histology types included)						
B vs. pT1	0.30	0.15	7.5e^−3^	−0.21	1.00	0.30
B vs. pT2	0.75	0.64	5.4e^−7^	−0.37	1.00	0.90
B vs. pT3	0.37	0.12	4.4e^−2^	−0.24	0.98	0.40
B vs. pT4	0.97	0.67	4.5e^−2^	−0.41	1.00	1.00
Benign vs RCC (ccRCC only)						
B vs. pT1	0.42	0.27	4.3e^−3^	−0.25	0.89	0.70
B vs. pT2	0.96	0.82	3.3e^−4^	−0.65	1.00	1.00
B vs. pT3	0.48	0.32	1.0e^−3^	−0.32	0.93	0.80
B vs. pT4	0.82	0.65	8.6e^−3^	−0.42	1.00	1.00
Histology Comparisons						
ONC vs. Chromophobe	0.88	0.77	3.0e^−2^	−0.51	1.00	1.00
ONC vs. ccRCC	0.32	0.20	4.8e^−4^	−0.24	1.00	0.20

Separation shown include between all histology subtypes and benign, ccRCC with benign lesions and between RCC histology subtypes from orthogonal partial least squares discriminant analysis (OPLS-DA) models. B: benign, ONC: oncocytomas, ccRCC: clear cell renal cell carcinoma.

**Table 3 metabolites-09-00155-t003:** Differential metabolites linked to renal cell carcinoma in ^1^H NMR analysis of urine.

Comparison	Increased in Cancer Relative to Benign	Decreased in Cancer Relative to Benign Samples
B vs. pT1	O-acetylcarnitine	Glycine
	Carnitine	Citrate
B vs. pT2	Gluconate	Citrate
	Carnitine	Creatinine
	-	Glycine
	-	Propylene Glycol
B vs. pT3	Carnitine	Pyridoxine
	O-acetylcarnitine	Adipate
	O-cresol	Citrate
	Methylhistidine	Glycine
B vs. pT4	Histamine	Citrate
	O-acetylcarnitine	Succinate
	Taurine	Glycine
	Carnitine	Glycerol
	5-aminolevulinate	-
	Carnitine	-
	Methionine	-
ONC vs. Chromophobe	Trans-aconitate	Citrate
	Succinate	-
	Methylhistidine	-
	Carnitine	-
ONC vs. ccRCC	Tartrate	1-methylnicotinamide
	Trans-aconitate	Glycine
	Histamine	2-hydroxyisobutyrate
	Carnitine	Citrate

Significant metabolites shown were selected based on VIP > 1.

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
