# Peer review of "Urinary Metabolomics Validates Metabolic Differentiation Between Renal Cell Carcinoma Stages and Reveals a Unique Metabolic Profile for Oncocytomas"

_metabolites, 2019, doi:10.3390/metabo9080155_

Round 1

Reviewer 1 Report

In this study Falegan et al. validate previously described urinary metabolic signature in a larger RCC patients cohort and provide a novel metabolic signature for oncocytoma.

The findings are interesting. I have some comments:

reference 1 should be updated

It could be interesting to evaluate the  association between metabolic features / signature and clinical outcomes (cancer specific survival and/or progression free survival) 

RCC is characterized by a reprogramming of energetic metabolism. In particular the metabolic flux through glycolysis is partitioned (PMID: 30983433), and mitochondrial bioenergetics and OxPhox are impaired (PMID: 30538212). These findings should be discussed and integrated with study results.

Reviewer 2 Report

This is an interesting manuscript which is worth to be published after major revision.

The main concern is about group size. Only 10 benign samples compared to for eg. 59 pT1 RCC samples. Why there are so unequal groups and what can be done with this?

Table 1 - Second column name should be "Number of samples" not "Number of sample"; 

Table 1- Third column name is unclear "Age at surgery range"?;

line 175- there should be "1-methylnicotinamide" instead of "1-methynicotinamide"

Figure 1, 2 and 3- Please check if the legends of x-axes of OPLS-DA models should not be "Sample number"? (instead of "Number of samples").

Table 3 is difficult for interpretation eg. which p-value is related to which comparison and metabolites.

OPLS-DA models were built on selected metabolites based on VIP value greater than 1. Did the authors check models built on all 73 metabolites?

Why some of the OPLS-DA models are built using only one component and others are built on more than one? In SIMCA software there is a possibility to add manually components to the model. Were models checked after the addition of more components? Was the separation between samples also observed?

The statistically significant models were confirmed by CV-ANOVA p-value greater than 0.05? Please see line 296

The statistically significant metabolites should be selected based on adjusted p-value (FDR or Bonferroni method). Was any other statistical method used for metabolites selection?

Was mannitol presence checked in samples during data processing? It can disturb spectra interpretation.

Was a comparison of benign samples vs all RCC samples performed?

How was the quality of the obtained spectra checked?

Round 2

Reviewer 2 Report

Thank you for all the corrections that you have done in the last version of the manuscript. Before final acceptance, I would recommend adding the score scatterplot showing the comparison Benign vs. all RCC samples (all histology types) to the supplementary materials. Also, I am not convinced by the explanation in Response 10. The best solution, in my opinion, is to use the FDR or Bonferroni methods and eventually based on experiments convince that this solution which is in the manuscript is beneficial or equally good. 
